# Enhancing RDX Thermal Decomposition in Al@RDX Composites with Co Transition Metal Interfacial Layer

Su-Lan Yang [1], Kan Xie [1,*], Jing Wang [1], Bingchen An [1], Bin Tian [1], Hongqi Nie [2], Jie-Yao Lyu [2] and Qi-Long Yan [2]

[1] Aerospace College, University of Electronic Science and Technology of China, Chengdu 611731, China; yangsl@uestc.edu.cn (S.-L.Y.); 202322100519@std.uestc.edu.cn (J.W.); 202321100509@std.uestc.edu.cn (B.A.)
[2] Science and Technology on Combustion, Internal Flow and Thermo-Structure Laboratory, Northwestern Polytechnical University, Xi'an 710072, China
[*] Correspondence: xiekan@uestc.edu.cn

**Abstract:** In this study, an Al/Co@RDX composite was meticulously prepared through a combination of planetary high-energy ball-milling and a spray-drying technique. The thermal reactivity of these Al/Co@RDX composites was comprehensively investigated and compared using the TG/DSC technique. It is shown that the initial decomposition temperature of RDX in the DSC curve was decreased by 26.3 °C in the presence of Al/Co, which could be attributed to the nano-sized Co transition metal catalyzing the decomposition reaction of nitrogen oxides in RDX decomposition products. The decomposition peak temperature of RDX and the heat released by the thermal decomposition of RDX in the Al/Co@RDX composite were decreased by 26.3 °C and increased by 74.5 J·g$^{-1}$, respectively, in comparison with those of pure RDX. The types of major gaseous products released from Al/Co@RDX were found to be identical to those of pure RDX, encompassing $N_2O$, $CH_2O$, $CO_2$ and HCN. However, the concentrations of those gaseous products for Al/Co@RDX were higher than those observed for pure RDX, which may owe to the fact that the Al/Co composite can interact with the –$CH_2$ and –$NO_2$ within RDX molecules, which leads to the weakening of the C-N and N-N bonds. In addition, the decomposition of RDX in the Al/Co@RDX composite was observed as a one-step process with an apparent activation energy ($E_a$) of 115.6 kJ·cm$^{-3}$. The decomposition mechanism of the RDX in the Al/Co@RDX composite was identified to follow the chain scission model (L2), whereas the two-step decomposition physical models observed for pure RDX were found to closely resemble the L2 and autocatalytic models.

**Keywords:** RDX; thermal reactivity; activation energy; Al/Co; gaseous products; decomposition mechanism

## 1. Introduction

Aluminum (Al) powder, the primary fuel component in solid propellants, exhibits considerable potential for enhancing the energetic performance of solid propellants [1–3]. However, traditional Al powders, particularly those with micron-sized particles, suffer from prolonged ignition delay owing to the presence of an Al oxide layer on their surface. This issue leads to the formation of sizable Al droplets during propellant combustion, resulting in a lower combustion efficiency and thereby restricting their application in propellants [4–8].

To address these challenges, extensive research has been globally conducted. The modification and enhancement of Al powder with various metallic elements have been found to significantly improve combustion efficiency and reduce the particle size of condensed phase combustion products in the combustion of propellants. Noteworthy influences include alkali metals (e.g., Li) [9–11], alkaline earth metals (e.g., Mg) [12–14], and transition metals (e.g., Fe, Co, Ni, Ti) [15–17]. A composite modification of Al powder with metallic elements weakens the protectivity of the alumina shell on its surface, facilitating the faster diffusion of oxidizers towards the Al core.

Alkali metals are characterized by their low melting points, low density, and high reactivity. Among them, lithium (Li), a member of the alkali metal group, stands out for its impressive combustion heat of 43.1 MJ·kg$^{-1}$, which is 39.5% higher than that of Al. This significant advantage of Li can be leveraged to enhance the energy release levels of Al fuels. Moreover, the distinctive capability of Li to react with Al and form an alloy introduces an extra dimension of functionality. This alloy formation is not only beneficial in terms of energy release but also capable of inducing a micro-explosion effect, which further underscores the potential of Li in enhancing the performance of Al-based fuels [18]. The micro-explosion phenomenon is triggered by the substantial differences in boiling points among various fuel components. This leads to the rapid atomization of the Al, resulting in more complete combustion and smaller condensed combustion products. Research has shown that Li plays a crucial role in enhancing various aspects of solid propellant combustion. It helps in reducing the activation energy, enhancing the combustion performance and improving the combustion characteristics of propellants, significantly reducing the ignition time from 11 ms to 5 ms [19]. Moreover, incorporating an Al-20Li (20 wt.% Li) alloy into AP-based (61.5 wt.% AP) composite propellants demonstrates environmental benefits, reducing hydrochloric acid generation by over 95% and increasing the theoretical specific impulse by approximately 7 s [10,18,20].

Apart from alkali metal Li, alkaline earth metals, especially Mg, display pronounced effects on reducing the ignition temperature, enhancing the combustion performance, and inhibiting the agglomeration of Al powder. The considerably lower boiling point of Mg (1170 °C) compared with that of Al (2477 °C) facilitates rapid vaporization during combustion, inducing micro-explosions that disperse the alloy powder into finer particles, thereby augmenting the reactive interface area and effective mass diffusion rate. The incorporation of Mg into Al powder produces an Al-Mg alloy powder that effectively suppresses Al agglomeration, reduces the ignition temperature, increases the combustion wave temperature (to as high as 1417 °C) [21], and significantly improves the combustion rate of the propellant by nearly two times compared the Al powder [13].

Additionally, the incorporation of transition metals (TMs) yields a substantial enhancement in the reactivity of Al powder [22–24]. TMs exhibit the capability to engage in reactions with Al powder, promoting its ignition and combustion performance. Due to the exceptional catalytic properties and reactivity of nano-sized TMs, they are commonly employed as combustion catalysts in solid propellants [25]. The addition of 1–3% nano-sized Cu or Ni powder was proven to be effective in significantly reducing the decomposition temperature and combustion pressure index of a perchlorate/hydroxyl-terminated polybutadiene (AP/HTPB) composite propellant, concurrently elevating the apparent heat release and burning rate [26]. The surface modification of Al powder with nano-sized TMs in a core-shell structure has been shown to enhance resistance to oxidation and combustion efficiency [1,27,28]. Preliminary research findings indicate that employing a spray-drying method to encapsulate nano-sized transition metals on the surface of Al powder results in the formation of complete Al/TM composite particles. This powder, while preserving the advantages of Al powder, was found to significantly reduce the size of condensed combustion products (by more than 64.2%) and the pressure index of AP/HTPB propellants (by 24% within the 0.5–3 MPa range). These findings strongly suggest that the introduction of transition metals plays a crucial role in enhancing the ignition efficiency of Al powder on the combustion surface of propellants.

Previous research has shown that the coating of a transition metal on the surface of Al powder significantly promotes propellant combustion performance. Studying the interaction of their composite system is crucial for revealing the energy release patterns that directly affect the energy performance of solid propellants, especially for high-energy composite propellants containing RDX and Al powder as the crucial energy sources. To investigate the introduction of a transition metal interfacial layer and its impact on the thermal decomposition and kinetic mechanisms of RDX is of great significance. In this study, the transition metal Co was selected and used to form an interface layer to prepare

high-energy Al/Co@RDX fuel with the spray-drying technique. RDX was uniformly distributed and tightly combined on the surface of Al/Co. The effects of the Co interface layer on the thermal decomposition and kinetic behaviors were systematically investigated with differential scanning calorimetry (DSC) and non-isothermal kinetic methods. The thermal decomposition mechanism of RDX was analyzed with the FTIR technique.

## 2. Experiment

### 2.1. Materials

Micro-sized spherical Al powder (μ-Al) with an average particle size of 5 μm and a purity greater than 99.9%, nano-sized spherical cobalt powder (n-Co, purity > 99.9%) averaging 500 nm in particle size, and dimethyl sulphoxide (DMSO, AR) were purchased from the Sigma-Aldrich company (Shanghai, China). Additionally, hexahydro-1,3,5-trinitro-1,3,5-trazine (RDX, ≥99.90%) was supplied by the Xi'an Modern Chemistry Research Institute (Xi'an, China). All materials were utilized as received without further purification processes.

### 2.2. Preparation Methods

The atomic ratio of Al to Co is 1:1 based on the optimal energy characteristics reported in the literature [29]. The optimal content of RDX in the Al/Co@RDX composite was determined with a Gaussian fitting analysis aimed to optimize the energy performance of Al/Co@RDX with different RDX contents. The results revealed that when the RDX content was 29.8%, Al/Co@RDX exhibited the highest energy-releasing performance, with an explosion heat value of 12,609 $J \cdot cm^{-3}$ (Table S1 and Figure S1). The Al/Co@RDX composite was prepared based on this formulation (Al:Co:RDX = 22.1:48.1:29.8). Figure 1 portrays the schematic diagram of its preparation process, which involved the preparation of Al/Co followed by the incorporation of RDX.

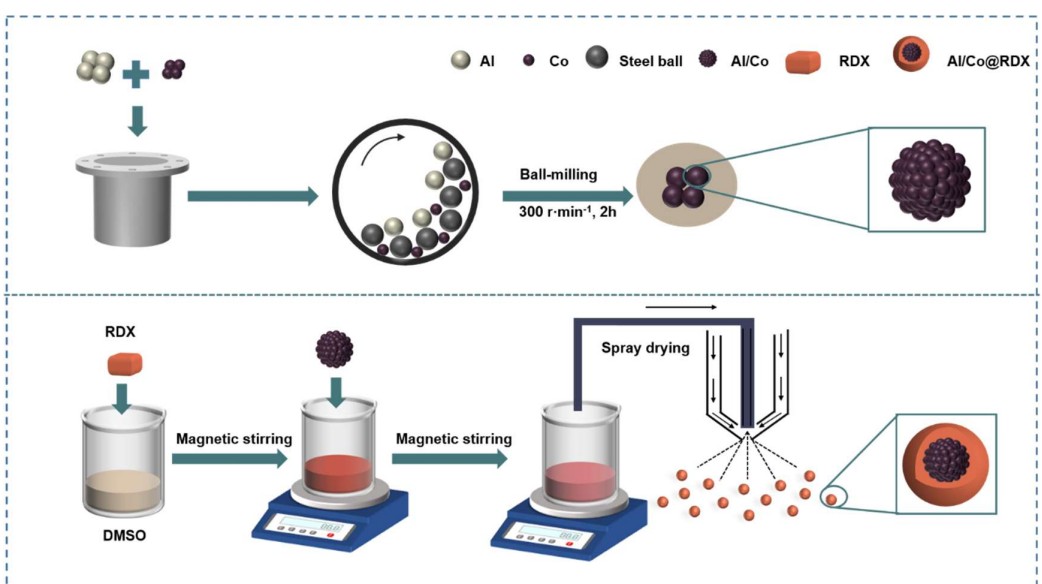

**Figure 1.** Schematic illustration of the preparation process of the Al/Co@RDX composite.

(i) The Al/Co nanocomposites were prepared by using a planetary high-energy ball-miller facility (XQW-2-DW, Changsha, China). More specific details regarding the preparation process of the Al/Co composite can be found in the recently published articles by our research group [30].

(ii) Afterwards, 2.98 mg of RDX was dissolved in 50 mL of a DMSO solvent and then subjected to violent magnetic stirring for approximately 30 min. Next, the Al/Co compound powder obtained from the high-energy ball-milling was dispersed in the abovementioned solution, followed by 20 min of ultrasound treatment and 1 h of

magnetic stirring at room temperature. Subsequently, the uniformly mixed precursor solution underwent the spray-drying process. The used spray-drying parameters were as follows: the diameter of the feed well was 1 mm, the fluid flow rate was 3 mL min$^{-1}$, and the inlet temperature was kept at 130 °C.

### 2.3. Characterization Technique

The morphologies of the raw materials and corresponding composites were characterized using field-emission scanning electron microscopy (FESEM, Hitachi 4800S, Tokyo, Japan). The composition of samples was tested with X-ray diffraction (XRD) with a Cu target (wavelength of 0.154 nm) covering a scanning range of 10–80°. The thermal decomposition behavior of RDX was analyzed with differential scanning calorimetry and thermal gravimetric (DSC/TG, STA449-F5, Netzsch, Germany) analyses at constant heating rates of 5 °C·min$^{-1}$, 10 °C·min$^{-1}$, 15·°C·min$^{-1}$ and 20 °C·min$^{-1}$ under Ar flow from 50 °C to 500 °C. Kinetic parameters for the thermal decomposition of RDX such as activation energy, pre-exponential factor and physical model were obtained with non-isothermal dynamics methods. The gaseous products resulting from the thermal decomposition of RDX were detected with Fourier transform infrared (FTIR, Bruker, Billerica, MA, USA) spectra, and the ATR spectra were collected at a resolution of 6 cm$^{-1}$ with a wavenumber range from 600 to 4000 cm$^{-1}$.

## 3. Results and Discussions

### 3.1. Morphology and Composition of Composites

Figure 2 vividly illustrates the morphology of the raw materials and the resultant composites. The Al powder exhibited a smooth, uniform spherical structure with an average particle diameter hovering around 5 μm (Figure 2a), while Co was discerned as a notably finer powder (as depicted in Figure 2b). Significant changes in the microstructure were evident after the high-energy ball-milling process, as the surface of the Al particles in the Al/Co mixture was evenly covered with numerous fine Co particles, although the larger particle structure of the Al itself remained mostly unchanged (Figure 2c). Subsequently, with the aid of the spray-drying process, RDX was uniformly dispersed over the surface of Al/Co, resulting in a unique honeycomb-like structure (Figure 2d). The formation of this structure could significantly impact the properties of the composite material, especially in terms of thermal conductivity and reaction rates.

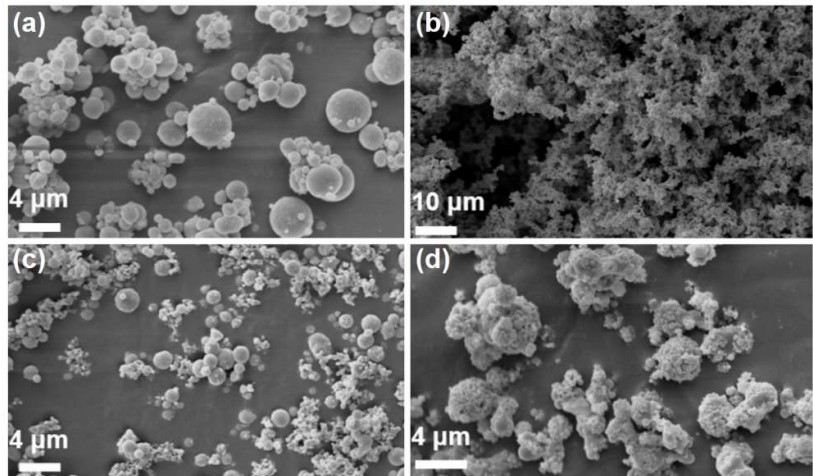

**Figure 2.** SEM micrographs of the powders involved in this study: (**a**) Al, (**b**) Co, (**c**) Al/Co and (**d**) Al/Co@RDX.

To further investigate the composition of the samples obtained with the aforementioned preparation methods, X-ray diffraction (XRD) analysis was conducted; the results are illustrated in Figure 3. The primary diffraction peaks of the raw material Al were

situated at 38.5°, 65.1° and 78.2°, while those of Co were located at 41.5°, 44.3°, 47.5° and 75.8°. The main diffraction peaks of Al/Co were aligned with those of the original Al and Co, signifying that the crystalline structure of Al/Co remained unchanged during the coating processes.

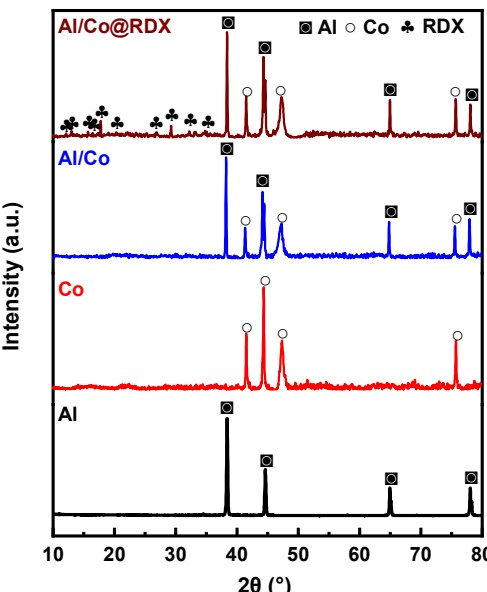

**Figure 3.** The XRD diffraction peaks of the raw materials and the Al/Co@RDX composite.

The XRD pattern of Al/Co@RDX shows that besides the diffraction peaks of Al and Co, there were additional diffraction peaks that appeared to correspond to RDX. This signifies the successful coating of RDX on the surface of Al/Co without inducing any noticeable phase transformation. After applying the spray-drying technique, a noticeable reduction in the intensity of the RDX diffraction peaks was observed. This phenomenon is primarily attributed to the diminution in RDX particle size. The irregular and periodic spatial distribution of RDX contributed to the weakening of the diffraction peak intensities.

### 3.2. Thermal Decomposition Behavior of RDX

The differential scanning calorimetry (DSC) thermal analysis technique with a heating rate of 10 °C·min$^{-1}$ was employed to investigate the thermal properties of RDX, and the corresponding DSC curves are presented in Figure 4. It is evident that the thermal decomposition behavior of pure RDX comprises two processes, as indicated by endothermic peaks at the melting temperature of 207.7 °C and a decomposition peak at 245.8 °C (Figure 4a), which is similar to values reported in the literature [31–34]. The initial peak temperature and heat of release for the decomposition process of RDX were recorded as 227.3 °C and 939.0 J·g$^{-1}$, respectively.

The addition of Al had a negligible impact on the melting and thermal decomposition processes of RDX, but it reduced the heat of release for RDX from 939.0 J·g$^{-1}$ to 763.6 J·g$^{-1}$ (Table 1). Conversely, in the case of Al/Co@RDX, the DSC curve for RDX shows no apparent solid-phase melting process, as it proceeded directly to thermal decomposition. In the presence of Al/Co, the initial decomposition temperature of RDX was decreased by 26.3 °C, the decomposition peak temperature was reduced by 32.5 °C, the exothermic peak became sharp, and the heat of release was increased by 74.5 J·g$^{-1}$ compared with that of pure RDX. This indicates that Al/Co facilitates the faster thermal decomposition of RDX, concentrating the exothermic reaction and highlighting a stronger catalytic effect compared with that of Al alone.

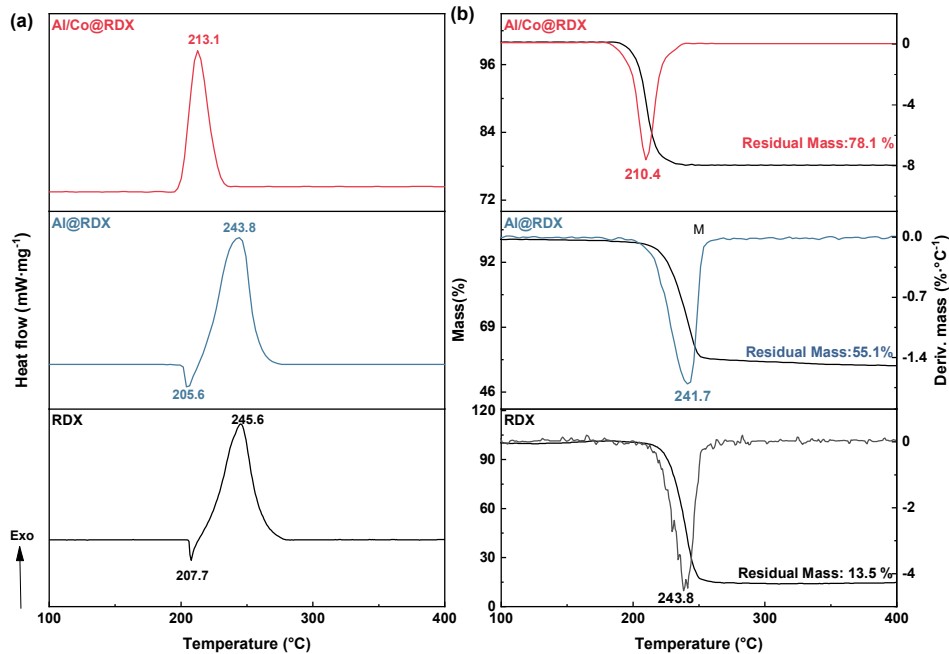

**Figure 4.** The (**a**) DSC and (**b**) TG–DTG curves for the studied Al/Co-based composites.

**Table 1.** The exothermic peak parameters of the DSC curves for the studied composites.

| Sample | $T_i$ (°C) | $T_p$ (°C) | $T_e$ (°C) | Width (°C) | $\Delta H$ (J·g$^{-1}$) |
|---|---|---|---|---|---|
| RDX | 227.3 | 245.6 | 256.9 | 23.6 | 939.0 |
| Al@RDX | 223.6 | 243.8 | 253.6 | 27.8 | 763.6 |
| Al/Co@RDX | 201.0 | 213.1 | 223.5 | 16.1 | 1113.5 |

The TG–DTG curves analysis reveals that both RDX and the involved composites exhibited a single weight loss process within the temperature range of 100–400 °C, which corresponded to the thermal decomposition of RDX in the samples. Compared with pure RDX, the presence of Al/Co resulted in decreases of 17.0 °C and 29.6 °C (Table 2) in the initial and end decomposition temperatures, respectively, of the RDX weight loss process. Additionally, the uncontrolled decomposition reaction temperature was decreased by 21.7 °C and the maximum weight loss rate was recorded at 23.58%·min$^{-1}$, as observed at a peak temperature of 210.4 °C. This indicates that the initial weight loss temperature, uncontrolled decomposition reaction temperature, and weight loss peak temperature of RDX were all slightly reduced due to the presence the transitional metal interfacial layer.

**Table 2.** The exothermic peak parameters of the TG–DTG curves for the studied composites.

| Sample | TG | | | | DTG | |
|---|---|---|---|---|---|---|
| | $T_i$ (°C) | $T_o$ (°C) | $T_e$ (°C) | Mass Loss (%) | $T_p$ (°C) | $L_{max}$ (%·min$^{-1}$) |
| RDX | 206.0 | 224.4 | 259.1 | 86.5 | 243.8 | −39.41 |
| Al@RDX | 204.3 | 217.5 | 257.0 | 44.9 | 241.7 | −17.20 |
| Al/Co@RDX | 189.0 | 202.7 | 229.5 | 21.9 | 210.4 | −23.58 |

In the Al/Co composite, the presence of a nano-sized Co transition metal introduces numerous twinning and void defects. These defects enable the removal of NO$_2$ and H from RDX molecules, leading to a weakening effect of the C-N and N-N bonds in RDX molecules and thereby promoting the thermal decomposition of RDX. Additionally, the nano-sized transition metal catalyzes the decomposition reaction of nitrogen oxides in the decomposition products of RDX, which further promotes the decomposition of RDX.

### 3.3. Non-Isothermal Dynamics Analysis of RDX Decomposition

The thermal decomposition behaviors of RDX, Al@RDX, and Al/Co@RDX under heating rates of 5, 10, 15, and 20 °C·min$^{-1}$ were investigated using differential scanning calorimetry (DSC). The respective DSC curves are depicted in Figure 5.

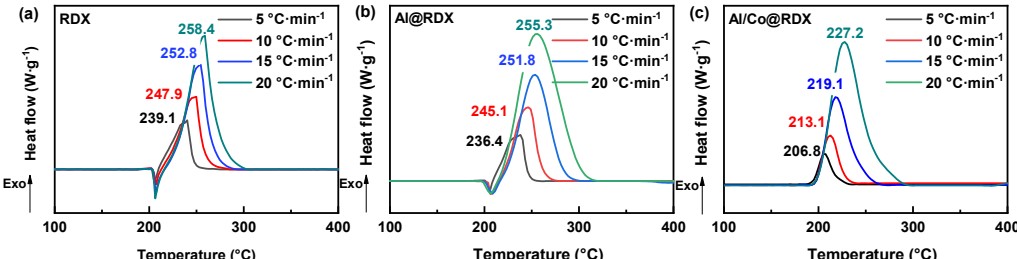

**Figure 5.** The DSC curves for (**a**) RDX, (**b**) Al@RDX and (**c**) Al/Co@RDX at different heating rates.

The decomposition peak temperatures of pure RDX were observed at 239.1, 247.9, 252.8, and 258.4 °C, respectively, at the elevated heating rates (Figure 5a). For Al@RDX, the corresponding decomposition peak temperatures were slightly lower, measured at 236.4, 245.1, 251.8, and 255.3 °C, respectively (Figure 5b). Notably, in the case of Al/Co@RDX, the decomposition peak temperatures were significantly reduced, registering at 206.8, 213.1, 219.1, and 227.2 °C, respectively, at the different heating rates. The considerable decreases in decomposition peak temperatures observed in Al/Co@RDX compared with both pure RDX and Al@RDX at all heating rates strongly suggest a pronounced catalytic effect of Al/Co on the thermal decomposition of RDX. This impact may be attributed to the presence of a transition metal interface layer and the interaction between Al and the transition metal, which promotes the establishment of more effective reaction pathways.

### 3.3.1. Kinetic Parameters Assessed with Kissinger Method

The reaction kinetics of RDX thermal decomposition were determined using the Kissinger method based on the DSC results obtained at various heating rates. Since the correlation coefficient of the one-step thermal decomposition of RDX was found to be 0.9543 (Figure S2a), which is considerably lower than the typically required value of 0.99 for precise kinetic evaluations, we employed the deconvolution method to separate the overlapping peaks of RDX [35]. Studies have indicated that the Fraser–Suzuki (FS) function is capable of fitting asymmetric differential curves. Therefore, we applied the FS equation to perform the peak-fitting analysis of the RDX decomposition process, which could effectively separate the overlapping reaction pathways. The two-step decomposition peaks were obtained after fitting the overlapping peaks of RDX (Figure S2b).

The impact of Al and Al/Co on the decomposition kinetics of RDX was comparatively analyzed. The calculated kinetic parameters of the activation energy ($E_a$), pre-exponential factor ($A$), and correlation coefficient ($r$) are summarized in Table 3. The calculated $E_a$ values for the two-step decomposition of RDX were 115.9 kJ·mol$^{-1}$ and 156.0 kJ·mol$^{-1}$. Notably, the $E_a$ values obtained for the two-step decomposition of RDX in the Al@RDX composite were decreased by 15.6 kJ mol$^{-1}$ and 55.7 kJ mol$^{-1}$, respectively, compared with those calculated for pure RDX. In case of Al/Co@RDX, the two-step decomposition reaction was merged into one step in the presence of Al/Co, indicating the high catalytic activity of Al/Co in the decomposition of RDX.

**Table 3.** Comparison of the kinetic parameters obtained with the Kissinger method for the thermal decomposition of RDX coated by Al and Al/Co.

| Sample | $E_a$/kJ mol$^{-1}$ | log A/s$^{-1}$ | r |
|---|---|---|---|
| RDX—1st | 115.9 | 6.12 | 0.9992 |
| RDX—2nd | 156.0 | 9.91 | 0.9995 |
| Al@RDX—1st | 100.3 | 4.41 | 0.9854 |
| Al@RDX—2nd | 75.1 | 1.7 | 0.9605 |
| Al/Co@RDX | 115.6 | 6.71 | 0.9991 |

3.3.2. The Dependence of $E_a$ on Conversion Degree Calculated with Friedman Method

The evolution of $E_a$ as a function of the conversion degree (*a*) was successfully determined using the Friedman method. The changing trends of RDX decomposition $E_a$ for RDX and its corresponding composites at different conversion degrees are illustrated in Figure 6.

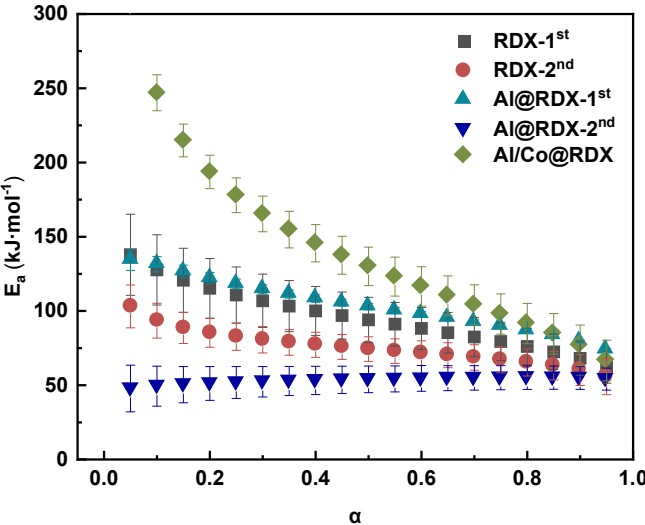

**Figure 6.** Activation energies vary as a function of the conversion degree obtained for both RDX and the corresponding RDX composites.

Figure 6 clearly shows that the activation energies obtained for the two-step decomposition of pure RDX decreased with increases in *a* in the range of 0 < *a* < 0.95. Under the effect of Al, the changing trend of the first-step $E_a$ of RDX decomposition was similar to that of pure RDX, while the second-step $E_a$ was significantly lower than that of pure RDX. The $E_a$ increased with *a* in the range of 0 < *a* < 0.45, but when *a* exceeded 0.5, the variation in $E_a$ became less noticeable and tended to be stabilized. In comparison, for Al/Co@RDX, the calculated $E_a$ of the one-step decomposition of RDX more significantly decreased within the range of 0 < *a* < 0.95, approaching the value of pure RDX at *a* = 0.95. This indicates that the addition of a Co transition metal interface strongly promotes the decomposition of RDX.

3.3.3. Physical Model Analyzed with a Combined Kinetic Method

To further investigate the intrinsic mechanism of RDX decomposition under the influence of Al and Al/Co, we employed a combined kinetics approach (CKA) to calculate a physical model of RDX decomposition. As illustrated in Figure 7, the physical model curve for RDX decomposition is represented by scattered points and the ideal model curve is depicted by a solid line.

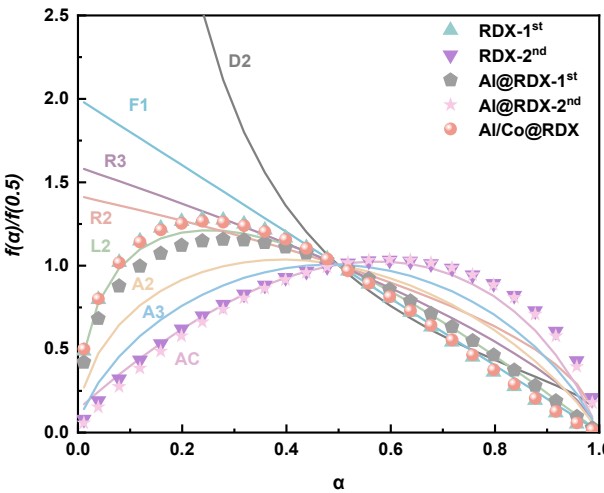

**Figure 7.** Comparison between the normalized curves of the physical RDX decomposition model and the ideal model. Notes: first-order reaction (F1), the so-called unimolecular decay law, in which random nucleation is followed by the instantaneous growth of nuclei; phase boundary-controlled reaction (contracting volume, R3); phase boundary-controlled reaction (contracting area, R2); chain scission model (L2); random nucleation, followed by the two- (A2) and three-dimensional (A3) growth of nuclei though different nucleation and nucleus growth models; autocatalytic model (AC).

It can be inferred from Figure 7 that the two-step decomposition process of RDX in the physical model corresponds to the chain cleavage model (L2) and autocatalytic model (AC). This suggests that the substances generated in the first step of RDX decomposition catalyze the second step reaction, thereby promoting the continuous progression of the entire reaction process. Interestingly, the encapsulation of RDX on the surface of the Al powder did not result in a significant alteration of its kinetic model. This indicates that the influence of Al powder on the physical model of RDX decomposition is not prominent. However, under the synergistic influence of Al/Co, the essentially one-step decomposition physical model of RDX was found to conform to the chain cleavage type. This implies that the combined action of Al and Co significantly affects the RDX decomposition process, leading to distinctive features in its physical model compared with Al powder alone.

### 3.4. Characterization of Gaseous Products Generated during the Thermal Decomposition of RDX

Through the use of a Fourier transform infrared spectrometer (FTIR), we conducted a comprehensive investigation into the gaseous decomposition products generated during the thermal decomposition of both pure RDX and Al/Co@RDX systems. The gaseous products formed during the decomposition of RDX in different composite systems were recorded in the FTIR spectra, as illustrated in Figure 8.

From the infrared spectra presented in Figure 8, it is evident that the thermal decomposition of both RDX and Al/Co@RDX yielded gases such as $N_2O$, $CH_2O$, $CO_2$ and HCN. The characteristic absorption peaks for these gases were observed at 2201 $cm^{-1}$ and 1629 $cm^{-1}$ for $N_2O$, from 2280 to 2350 $cm^{-1}$ for $CH_2O$, 2358 $cm^{-1}$ for $CO_2$, and 713 $cm^{-1}$ for HCN [36]. Notably, $N_2O$ exhibited the most prominent absorption peak, indicating a higher production rate of $N_2O$ during the decomposition process.

For an in-depth analysis, we investigated the absorbance intensities of various gaseous products of RDX under the influence of Al/Co at different temperatures and compared them with those of pure RDX, as depicted in Figure 9. Noticeably, the absorbance intensities of RDX under the influence of Al/Co gaseous products at the same temperature were significantly higher than those recorded for pure RDX. This intriguing outcome can be attributed to the unique characteristics of the Al/Co composite, including its small particle size and larger surface area. Due to the small particle size and larger surface area of the Al/Co composite, a considerable number of surface atoms remain in an unsaturated state,

creating distinctive surface effects. Hence, the Al/Co composite can interact with the -$CH_2$ and -$NO_2$ within RDX molecules, leading to the weakening of the C-N and N-N bonds and consequently promoting the thermal decomposition process of RDX. This result suggests that the Al/Co interfacial layer plays a significant role in enhancing the decomposition of RDX, leading to the generation of a greater quantity of gaseous products.

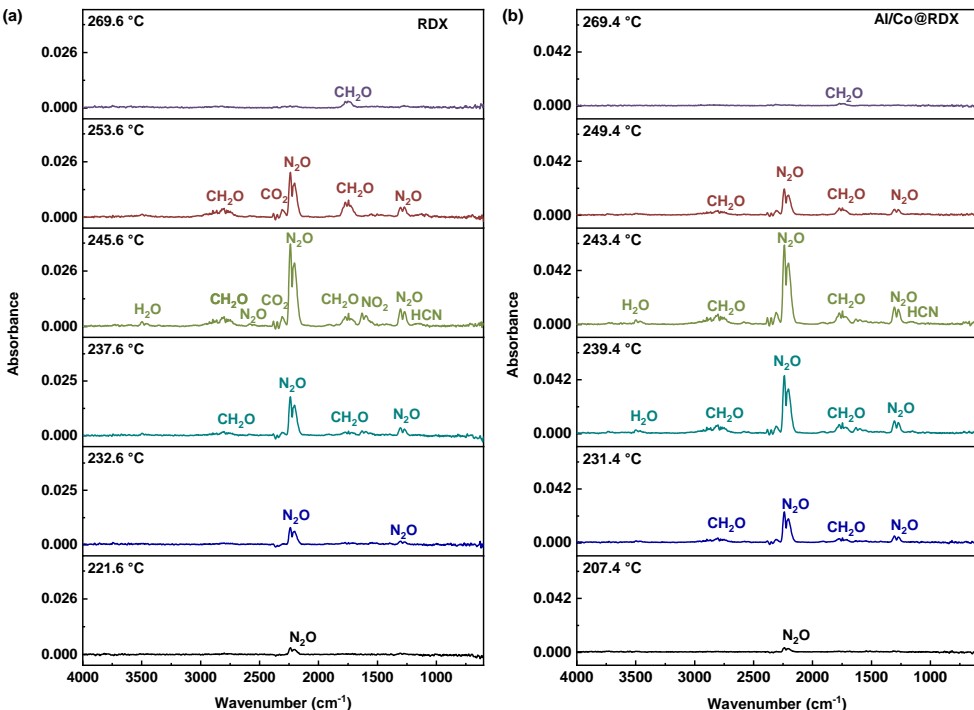

**Figure 8.** Compiled FTIR spectra recorded for (**a**) pure RDX and (**b**) Al/Co@RDX.

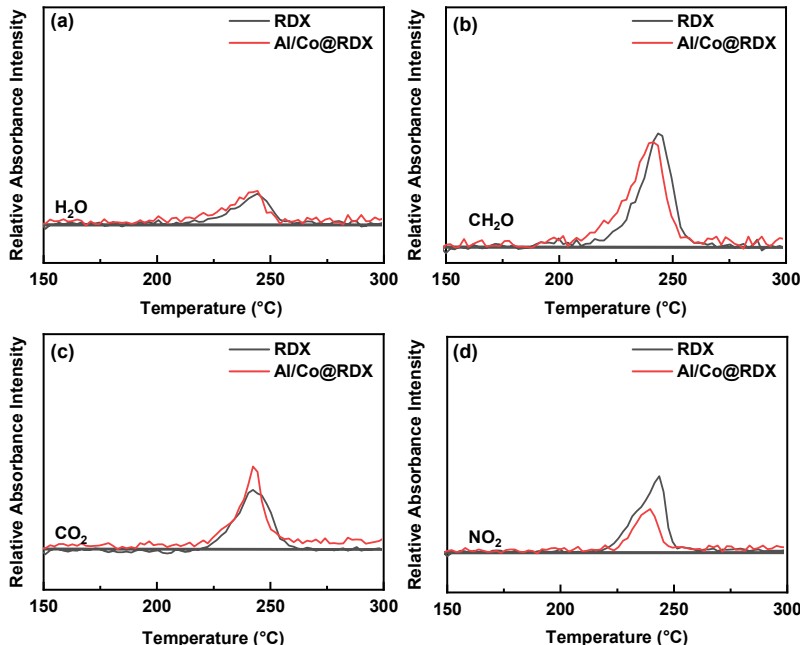

**Figure 9.** Comparison of the temperature-dependent relative gas absorption intensity between RDX and Al/Co@RDX for: (**a**) $H_2O$, (**b**) $CH_2O$, (**c**) $CO_2$ and (**d**) $NO_2$.

## 4. Conclusions

In this work, a core-shell-structured Al/Co@RDX was successfully prepared through a combination of planetary high-energy ball-milling and a spray-drying technique. The

transition metal Co was strategically utilized as an interface modifier that was capable of realizing precise control over the thermal decomposition performance of RDX. The effects of Co interface materials on the thermal decomposition and reaction kinetics of RDX were comprehensively investigated. The principal findings and conclusions of this research are succinctly summarized as follows:

(1) The optimal content of RDX in the Al/Co@RDX composite is 29.8% RDX, with a high energy value of 12,609 J·cm$^{-3}$. Compared with pure RDX, the introduction of the Al/Co composite into RDX was found to significantly enhance its thermal decomposition properties. The decomposition peak temperature decreased by 32.5 °C, and the decomposition heat release was increased by 74.5 J·g$^{-1}$.

(2) The Kissinger method was employed to discern kinetic parameters, revealing a decrease in the activation energy for Al@RDX and a distinctive merging of decomposition steps in Al/Co@RDX. A holistic evaluation using a combined kinetics approach uncovered unique features in the physical model of RDX decomposition in the presence of Al/Co, emphasizing its significant influence on the Co interface materials.

This study contributes valuable insights into the intricate interplay of Al/Co modification on RDX, offering implications for the development of high-performance solid propellants in aerospace applications.

**Supplementary Materials:** The following supporting information can be downloaded at: https://www.mdpi.com/article/10.3390/aerospace11010081/s1, Table S1: Heat of reaction Al/Co/RDX composites with different oxidant contents measured by oxygen bomb calorimeter; Figure S1: Fitting curves for heat of reaction of Al/Co/RDX composites; Figure S2: The fitting curve of RDX decomposition reaction peaks includes overlapping reactions of one step (a) and two steps (b).

**Author Contributions:** Conceptualization, Q.-L.Y.; methodology, H.N.; Software, K.X.; Validation, S.-L.Y.; formal analysis: S.-L.Y. and J.W.; investigation, B.T. and J.-Y.L.; Resources, K.X.; data curation, S.-L.Y. and B.A.; writing—original draft preparation, S.-L.Y., J.W., B.A., B.T. and J.-Y.L.; writing—review and editing, K.X., H.N. and Q.-L.Y. All authors have read and agreed to the published version of the manuscript.

**Funding:** This research received no external funding.

**Data Availability Statement:** The data presented in this study are available on request from the corresponding author.

**Conflicts of Interest:** The authors declare no conflict of interest.

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
