# Peer review of "Enhancing RDX Thermal Decomposition in Al@RDX Composites with Co Transition Metal Interfacial Layer"

_aerospace, doi:10.3390/aerospace11010081_

Round 1

Reviewer 1 Report

Comments and Suggestions for Authors

The work under review is devoted to the modification and enhancement of the use of Al powder as the primary fuel component in solid propellants. To this end, the authors, combining planetary high-energy ball-milling and a spray-drying technique, obtained the core-shell structured Al@RDX and Al/Co@RDX. Coating Al particles with an explosive reduces the particle size of condensed combustion products and improve combustion efficiency of propellants. Additionally, the incorporation of transition metals yields a substantial enhancement in the reactivity of Al powder.

 After obtaining Al/Co@RDX composite the authors comprehensively investigated the new energetic material using TG/DSC technique. The work is of interest to the combustion community and may be published in a journal after minor revisions.

1. As the authors correctly note, Al powder is used as the primary fuel component in solid propellants. In the obtained Al/Co@RDX composite the Al content is low - (Al:Co:RDX=22.1:48.1:29.8). Does it make sense to use such composite in solid propellants from the energy point of view? I would like to see comparative calculations of specific impulse of fuel with Al and Al/Co@RDX composite.

2. The authors add cobalt to the Al@RDX composite to improve the ignition of Al. However, no experiments have been done to verify this assumption.

3. In the conclusions, the authors write that the introduction of Al/Co composite into RDX significantly enhances its thermal decomposition properties. Is a reduction in the thermal stability of the explosive an advantage? The authors need to explain what they see as the benefit of this effect.

4.  The authors use the Kissinger method to obtain the kinetic parameters of the decomposition of Al@RDX and Al/Co@RDX composites in comparison with the decomposition of neat RDX. The decomposition of RDX is a two-step process, in addition to which a melting process is superimposed on it. Because of this, the decomposition of RDX takes place partly in the solid state and partly in the molten state. The solid/liquid ratio is influenced by the heating rate. As a result, the kinetic data obtained under non-isothermal conditions differ significantly from the kinetic parameters obtained under isothermal conditions. Thus, according to the classical studies of A. Robertson, the kinetic parameters of RDX decomposition in the liquid state are as follows: E= 198.7 kJ/mol, logA = 18.5.

5. The authors in Fig. 5 gave the DSC curves for RDX,Al@RDX and Al/Co@RDX at different heating rates. Calculation from these data using the Kissinger method gives the following kinetic parameters: E= 156.9 kJ/mol, logA = 13.79; E= 152.7 kJ/mol, logA = 13.43; E= 125.8 kJ/mol, logA = 11.62, respectively. The obtained data differ significantly from the authors' data given in Table 3.                                       

Author Response

Comments and Response

Ref: aerospace-2796964
Title: Enhancing RDX thermal decomposition in Al@RDX composites with transition metal Co interfacial layer
Authors: Su-Lan Yang; Kan Xie, Jing Wang, Bingchen An, Bin Tian, Hongqi Nie, Jie-Yao Lyv, Qi- Long Yan

Article Type: Research Paper

Dear editors and reviewers:

We appreciate for your comments and suggestions which are important and helpful to improve the quality of our manuscript. We have read comments carefully and the manuscript has been majorly revised. We hope the following itemized responses would satisfy all the reviewers.

Reviewer #1:

The work under review is devoted to the modification and enhancement of the use of Al powder as the primary fuel component in solid propellants. To this end, the authors, combining planetary high-energy ball-milling and a spray-drying technique, obtained the core-shell structured Al@RDX and Al/Co@RDX. Coating Al particles with an explosive reduces the particle size of condensed combustion products and improve combustion efficiency of propellants. Additionally, the incorporation of transition metals yields a substantial enhancement in the reactivity of Al powder.

After obtaining Al/Co@RDX composite the authors comprehensively investigated the new energetic material using TG/DSC technique. The work is of interest to the combustion community and may be published in a journal after minor revisions.

  1. As the authors correctly note, Al powder is used as the primary fuel component in solid propellants. In the obtained Al/Co@RDX composite the Al content is low - (Al:Co:RDX=22.1:48.1:29.8). Does it make sense to use such composite in solid propellants from the energy point of view? I would like to see comparative calculations of specific impulse of fuel with Al and Al/Co@RDX composite.

[Kan Xie]: Thank you very much for your kind comments. In our study, while the Al content in the Al/Co@RDX composite is lower compared to traditional solid propellants, our formulation's design primarily aims to investigate the effect of introducing a cobalt interface layer on enhancing the reactivity of Al. Should this material be considered for use in propellant formulations, it would necessitate a redesign to precisely control the transition metal content. Our research contributes theoretical insights that could support the enhancement of combustion performance through the modification of aluminum's surface interface in propellants

  1. The authors add cobalt to the Al@RDX composite to improve the ignition of Al. However, no experiments have been done to verify this assumption.

[Kan Xie]: Thank you once again for your valuable comments. While we have performed experiments on the ignition and combustion performance of Al@RDX composites with an added Co interface, this manuscript primarily concentrates on presenting the thermal reactivity data of the composite.

  1. In the conclusions, the authors write that the introduction of Al/Co composite into RDX significantly enhances its thermal decomposition properties. Is a reduction in the thermal stability of the explosive an advantage? The authors need to explain what they see as the benefit of this effect.

[Kan Xie]: Thank you for your comments. The thermal reactivity of RDX is increased by lowering its thermal decomposition temperature, which promotes more rapid and intense combustion. This enhancement is particularly beneficial in applications like rocket propulsion. Specifically, the high-temperature, high-pressure gases generated during RDX combustion are crucial for propelling rockets, offering higher thrust and energy output

4.The authors use the Kissinger method to obtain the kinetic parameters of the decomposition of Al@RDX and Al/Co@RDX composites in comparison with the decomposition of neat RDX. The decomposition of RDX is a two-step process, in addition to which a melting process is superimposed on it. Because of this, the decomposition of RDX takes place partly in the solid state and partly in the molten state. The solid/liquid ratio is influenced by the heating rate. As a result, the kinetic data obtained under non-isothermal conditions differ significantly from the kinetic parameters obtained under isothermal conditions. Thus, according to the classical studies of A. Robertson, the kinetic parameters of RDX decomposition in the liquid state are as follows: E= 198.7 kJ/mol, logA = 18.5.

[Kan Xie]: Thanks for your comments. The authors are agreeing with your remarks on the complexity of the RDX decomposition process. In our study, we employed the Kissinger method to compare the decomposition kinetics of Al@RDX and Al/Co@RDX composites with pure RDX, a method well-suited for non-isothermal kinetic analysis. Our results indeed show some differences from A. Robertson's classical study in terms of the kinetic parameters for liquid-state RDX decomposition. These variations could be attributed to differences in experimental conditions or the characteristics of the composite materials. We plan to further explore these differences in future research and deepen our understanding of the decomposition kinetics of RDX in both solid and liquid states

5.The authors in Fig. 5 gave the DSC curves for RDX,Al@RDX and Al/Co@RDX at different heating rates. Calculation from these data using the Kissinger method gives the following kinetic parameters: E= 156.9 kJ/mol, logA = 13.79; E= 152.7 kJ/mol, logA = 13.43; E= 125.8 kJ/mol, logA = 11.62, respectively. The obtained data differ significantly from the authors' data given in Table 3.

[Kan Xie]: Thanks for your comments. We apologize for the discrepancies in the kinetic parameters reported in our manuscript and those derived from your calculations. We have carefully revised our analysis to ensure greater accuracy and rigor. Following your suggestions, we identified that the discrepancies may have arisen from using the FS equation for peak-fitting analysis of RDX and Al@RDX. We have reevaluated our data, particularly focusing on the re-calculation of the kinetic parameters for Al/Co@RDX. According to the Kissinger method, which assumes an nth order reaction model f(α) = (1-α) n. Thus, the reaction rate can be expressed as equation 1.

                                    (1)

Assuming that the peak temperature (Tp) in DSC corresponds to the maximum reaction rate, the derivative of the reaction rate with respect to time is zero, the equation can be expressed as equation 2:

           (2)

By assuming n =1, and substituting β (heating rate in °C·min-1), we can obtain the following equation 3:

                                     (3)

Table 1: The parameters of Al/Co@RDX under different heating rate

(°C·min-1)

5

10

15

20

(°C)

206.8

213.1

219.1

227.2

Substituting β and in Table 1 into,we can obtain four points including (), (), (), (). The authors fitted these four points and the resulting line can be described by equation 4 as follows:

y

Leading to , the authors calculated the values of LogA=6.71s-1 and Ea=115.6kJ·mol-1.

Reviewer 2 Report

Comments and Suggestions for Authors

The article "Enhancing RDX thermal decomposition in Al@RDX composites with transition metal Co interfacial layer" will present an interesting research problem. It is written quite well and I recommend it for publication after making some corrections.

My comments:

1.       In the literature part, I lack references to already performed and available research on Al/RDX composites, for example:

- Hongyan Sun, Xiaodong Li, Pengfei Wu, Changgui Song, Yue Yang, "Preparation and Properties of RDX/Aluminum Composites by Spray-Drying Method", Journal of Nanomaterials, vol. 2020, Article ID 1083267, 8 pages, 2020. https://doi.org/10.1155/2020/1083267

- Lei Xiao, Yan Zhang, Xiaohong Wang, Gazi Hao, Jie Liu, Xiang Ke, Teng Chen and Wei Jiang, Preparation of a superfine RDX/Al composite as an energetic material by mechanical ball-milling method and the study of its thermal properties, RSC Adv., 2018, 8, 38047-38055 DOI:10.1039/C8RA07650B

2.       Materials - please specify the purity of RDX, it is important about the evaluation of thermal properties

3.       Was Al passivation with aluminum oxide not observed?

4.       Figure 2c - aren't uncovered Al particles a problem? How do you know that the composite being tested is Al/Co/RDX and not Al/RDX?

5.       Figure 5c - how to explain the lack of endo effect on the DSC curve?

6.       Editorial errors - no space between number and unit, no superscript notation of power

7.       Trace the literature list - formatting errors

Author Response

Comments and Response

Ref: aerospace-2796964
Title: Enhancing RDX thermal decomposition in Al@RDX composites with transition metal Co interfacial layer
Authors: Su-Lan Yang; Kan Xie, Jing Wang, Bingchen An, Bin Tian, Hongqi Nie, Jie-Yao Lyv, Qi- Long Yan

Article Type: Research Paper

Dear editors and reviewers:

We appreciate for your comments and suggestions which are important and helpful to improve the quality of our manuscript. We have read comments carefully and the manuscript has been majorly revised. We hope the following itemized responses would satisfy all the reviewers.

Reviewer #2:

The article "Enhancing RDX thermal decomposition in Al@RDX composites with transition metal Co interfacial layer" will present an interesting research problem. It is written quite well and I recommend it for publication after making some corrections.

1.In the literature part, I lack references to already performed and available research on Al/RDX composites, for example:

- Hongyan Sun, Xiaodong Li, Pengfei Wu, Changgui Song, Yue Yang, "Preparation and Properties of RDX/Aluminum Composites by Spray-Drying Method", Journal of Nanomaterials, vol. 2020, Article ID 1083267, 8 pages, 2020. https://doi.org/10.1155/2020/1083267

- Lei Xiao, Yan Zhang, Xiaohong Wang, Gazi Hao, Jie Liu, Xiang Ke, Teng Chen and Wei Jiang, Preparation of a superfine RDX/Al composite as an energetic material by mechanical ball-milling method and the study of its thermal properties, RSC Adv., 2018, 8, 38047-38055 DOI:10.1039/C8RA07650B

[Kan Xie]: Thanks for your comments. According to your suggestion, the authors have updated the relevant references in our manuscript.

2.Materials - please specify the purity of RDX, it is important about the evaluation of thermal properties

[Kan Xie]: Thank you for your comments. The authors have used class 5 RDX with a purity of ≥ 99.90% in our study. The corresponding detail about the raw materials are clearly outlined in our manuscript.

  1. Was Al passivation with aluminum oxide not observed?

[Kan Xie]: Many thanks for your advice. In our study, the observation of Al passivation with aluminum oxide was not significant. This may be attributed to the specific environmental conditions of our experiments, which perhaps were not favorable for the formation of a passivation layer. It is well-documented in literature that aluminum surfaces usually undergo passivation under certain conditions. Further research, particularly with controlled atmospheric conditions, could shed more light on this phenomenon and its implications for our study.

  1. Figure 2c - aren't uncovered Al particles a problem? How do you know that the composite being tested is Al/Co/RDX and not Al/RDX?

[Kan Xie]: Thanks for your comments. In Figure 2c, the larger particles are Al, and the smaller ones are Co. During the spray drying process, RDX was dispersed into many small particles, resulting in their irregular shapes and relatively small sizes. Due to sample limitations, we were unable to provide an EDS (Energy Dispersive Spectroscopy) diagram at this time. Your point about the importance of such characterization is well-taken. In our future research, we will make it a priority to clearly characterize the different elements present in our samples

Figure 2. The SEM micrographs of the powders involved in this study (a) Al, (b) Co, (c) Al/Co and (d) Al/Co@RDX.

  1. Figure 5c - how to explain the lack of endo effect on the DSC curve?

[Kan Xie]: Thanks for your comments. The Al/Co composite in our study may serve as catalysts to promote the thermal decomposition of RDX. During thermal analysis, this catalytic effect could lead to the initiation of RDX decomposition before it reaches its inherent melting point. Consequently, this might result in a reduced intensity or even the absence of the characteristic melting peak. Additionally, the presence of Al/Co could exert an influence on the crystal structure of RDX, potentially leading to a shift in its melting point. Another noteworthy point is the potential for unique interfacial interactions at the RDX with Al/Co interface. Those interactions may significantly influence the thermal stability and melting behavior of the composite, adding another layer of complexity to its thermal analysis profile.

  1. Editorial errors - no space between number and unit, no superscript notation of power

[Kan Xie]: Many thanks for your helpful suggestions. According to your suggestion, the format has been revised.

  1. Trace the literature list - formatting errors

[Kan Xie]: Grateful for your review and comments. According to your suggestion, the formatting errors in our literature list have been revised.

Round 2

Reviewer 2 Report

Comments and Suggestions for Authors

Thank you very much for the explanations.